Identification of hub genes and small-molecule compounds in medulloblastoma by integrated bioinformatic analyses

Liu Zhendong
Zhang Ruotian
Sun Zhenying
Yao Jiawei
Yao Penglei
Chen Xin
Wang Xinzhuang
Gao Ming
Wan Jinzhao
Du Yiming
Zhao Shiguang guangsz@hotmail.com
Department of Neurosurgery, The First Affiliated Hospital of Harbin Medical University , Harbin , Heilongjiang Province , People’s Republic of China
Institute of Brain Science, Harbin Medical University , Harbin , Heilongjiang Province , People’s Republic of China
Abdullah Jafri
Electronic publication date: 2020 Apr 14
Publication date: 2020
Volume: 8
Electronic Location ID: e8670
Received 2019 Oct 8; Accepted 2020 Jan 30
Copyright: ©2020 Liu et al.
Copyright year: 2020
Copyright holder: Liu et al.
License: This is an open access article distributed under the terms of the Creative Commons Attribution License, which permits unrestricted use, distribution, reproduction and adaptation in any medium and for any purpose provided that it is properly attributed. For attribution, the original author(s), title, publication source (PeerJ) and either DOI or URL of the article must be cited.
License URL: https://creativecommons.org/licenses/by/4.0/

Keywords: Bioinformatical analysis, Differentially expressed genes, Medulloblastoma, Small-molecule drugs

Funding: Special Fund for Translational Research of Sino-Russia Medical Research Center in Harbin Medical University CR201410 CR201512 This study was supported by the Special Fund for Translational Research of Sino-Russia Medical Research Center in Harbin Medical University (CR201410, CR201512). The funders had no role in study design, data collection and analysis, decision to publish, or preparation of the manuscript.

==============================
Background

Medulloblastoma (MB) is the most common intracranial malignant tumor in children. The genes and pathways involved in the pathogenesis of MB are relatively unknown. We aimed to identify potential biomarkers and small-molecule drugs for MB.

Methods

Gene expression profile data sets were obtained from the Gene Expression Omnibus (GEO) database and the differentially expressed genes (DEGs) were identified using the Limma package in R. Functional annotation, and cell signaling pathway analysis of DEGs was carried out using DAVID and Kobas. A protein-protein interaction network was generated using STRING. Potential small-molecule drugs were identified using CMap.

Result

We identified 104 DEGs (29 upregulated; 75 downregulated). Gene ontology analysis showed enrichment in the mitotic cell cycle, cell cycle, spindle, and DNA binding. Cell signaling pathway analysis identified cell cycle, HIF-1 signaling pathway, and phospholipase D signaling pathway as key pathways. SYN1, CNTN2, FAIM2, MT3, and SH3GL2 were the prominent hub genes and their expression level were verified by RT-qPCR. Vorinostat, resveratrol, trichostatin A, pyrvinium, and prochlorperazine were identified as potential drugs for MB. The five hub genes may be targets for diagnosis and treatment of MB, and the small-molecule compounds are promising drugs for effective treatment of MB.

Conclusion

In this study we obtained five hub genes of MB, SYN1, CNTN2, FAIM2, MT3, and SH3GL2 were confirmed as hub genes. Meanwhile, Vorinostat, resveratrol, trichostatin A, pyrvinium, and prochlorperazine were identified as potential drugs for MB.

Introduction

Medulloblastoma (MB) is one of the most common intracranial malignant tumors in children. Standardized treatment options including maximal surgical resection, radiation therapy, and chemotherapy can improve overall five-year survival rates; however, the prognosis of patients with advanced disease is still unsatisfactory (Kim et al., 2010). Despite extensive studies on the pathogenesis and progression of MB, the pathophysiology of disease development is still unclear. The pathogenesis of MB is closely related to many factors such as mutations in genes, abnormalities in cellular immunity, and changes in environmental factors (Wang et al., 2018). A deep understanding of the changes in protein expression involved in the pathogenesis of MB is critical for the development of better treatment strategies.

Gene chip is a genetic testing technology that has been in use for more than a decade. The gene chip technology can rapidly detect changes in gene expression in the sample and is suitable for screening differentially expressed genes (DEGs) (Vogelstein et al., 2013). In recent years, gene chip technology has been widely used, and data from numerous microarray studies have been stored in free public databases. These databases provide valuable data for further research. So far, several gene chip data have been analyzed, and hundreds of different genes involved in the development of several central nervous system cancers such as gliomas have been identified (Xi et al., 2017; Zeng et al., 2018). However, most gene chip data is a mixture of both MB and glioma and does not accurately represent changes in MB, and therefore does not generate satisfactory results. There have been reports of microarray data to find DEGs in medulloblastoma (Shaabanpour Aghamaleki et al., 2019). However, the reliability of the analysis results was controversial due to the heterogeneity of the sample from a single cohort study. Therefore, most single-chip analyses are not sufficient to identify effective biomarkers in MB. To overcome this, we used data derived from four gene chips to identify effective biomarkers for MB. This method is more accurate and does not have the disadvantages associated with the single-chip analysis.

In this study, we downloaded four MB microarray datasets (GSE42656, GSE74195, GSE109401, and GSE50161) from free public Gene Expression Omnibus database (GEO). A total of 77 MB samples and 39 normal brain samples were included in this study. We identified the DEGs using the Limma package in R software (version 3.5.0) from the four gene expression profiles and subsequently, used the Venny online tool for further integrated analyses. We then employed the DAVID databases and Kobas online tool to identify the functions of the identified DEGs and the key cell signaling pathways involved. The network of protein-protein interaction (PPI) was generated using the STRING database. Finally, we used the CMap database to explore potential small-molecule compounds that can be used for treating MB.

Materials & Methods

Microarray data information

We downloaded the gene expression profile data for GSE42656, GSE74195, GSE109401, and GSE50161 from the GEO database (http://www.ncbi.nlm.nih.gov/geo). The GSE42656 data set was based on the GPL6947 Platforms (Illumina HumanHT−12 V3.0 expression beadchip) and contained nine MB samples and 16 normal brain samples (Henriquez et al., 2013). The GSE74195 data set was based on the GPL570 Platform (Affymetrix Human Genome U133 Plus 2.0 Array) and contained 27 MB and five normal brain tissue samples (De Groot et al., 2011). The GSE50161 data set was based on the GPL570 Platform (Affymetrix Human Genome U133 Plus 2.0 Array) and included 22 MB samples and 13 normal brain tissues (Griesinger et al., 2013). The GSE109401 data set was based on GPL16686 Platforms (Affymetrix Human Gene 2.0 ST Array (transcript (gene) version)) and included 19 medulloblastoma samples and five normal brain samples (Rivero-Hinojosa et al., 2018). We selected these four gene expression profiles for further integrated analyses to avoid racial differences and errors in individual experiments.

Identification of DEGs in medulloblastoma

The DEGs were identified based on the series matrix file using Limma package in R software (version 3.5.0) according to the cut-off standard (p < 0.05 and logFC >1) (Fang et al., 2017). The four sets of differential expression data were respectively divided into upregulated DEmRNAs and downregulated DEmRNAs and were uploaded to Venny (http://bioinfogp.cnb.csic.es/tools/venny/) for integrated analyses. A heat map was generated using the pheatmap package in R. The GSE74195 dataset was used as the reference to generate the heat map. The heatmap showed 29 upregulated DEGs, and 75 downregulated DEGs.

GO and KEGG enrichment analysis of DEGs

The DAVID online database and Kobas online tool were used for functional and pathway enrichment analyses (Huang da, Sherman & Lempicki, 2009). We uploaded the DEGs to DAVID for GO functional annotation analysis and Kobas for KEGG pathway enrichment analysis. P < 0.05 was used as the cut-off value.

Building a PPI network

The STRING public database, which provides PPI network analysis, can evaluate direct and indirect links between DEGs (https://string-db.org) (Franceschini et al., 2013). The DEGs were uploaded to STRING to obtain the PPI network using an Interaction score of 0.2. Cytoscape was used for visualizing the PPI network and for identifying the hub genes based on the degree of connectivity between DEGs.

Identification of small-molecule compounds for the treatment of MB

CMap, an online tool, can be based on the gene expression profile of a disease to mine potential therapeutic drugs (Lamb et al., 2006). We divided the DEGs of MB into upregulated DEGs and downregulated DEGs, and uploaded them to CMap to explore the drugs. P < 0.001 and Enrichment <-0.8 was used as the cut-off criteria. The 3D structures of the small-molecule compounds are available from PubChem (https://pubchem.ncbi.nlm.nih.gov/).

RNA isolation and reverse transcription quantitative polymerase chain reaction (RT-qPCR) analysis

Total RNA is extracted from in normal human brain tissue and medulloblastoma cell lines (Daoy and D283) using Tri-Reagent (Sigma, USA) according to the manufacturer’s instructions. NanoDrop One spectrophotometer (Thermo Fisher Scientific, USA) was used to evaluate the quality and quantity of the RNA. The Transcriptor First Stand cDNA Synthesis Kit (Roche, USA) was used to reverse transcribe total RNA into cDNA.The FastStart Universal SYBR Green Master (ROX) (Roche, Germany) and the QuantStudio software (Thermo Fisher Scientific, USA) were reserved for RT-qPCR based on the manufacturer ’s instructions.The GAPDH gene was used as an endogenous reference.

The primer sequences were as follows: 5′-GGACACGTGCTCAGAGATT-3′ (sense) and 5′-TCTACGATGAGCTGTTTGTCTTC-3′ (antisense) for SYN1, 5′-GGGGTGATGTTGCCCTGTAA-3′ (sense) and 5′-AGGTCTGAGGCATTGGTTCG-3′(antisense) for CNTN2, 5′-CTGATTCTCCTGACCGTCTTTAC-3′ (sense) and 5′-GAACTTGGTCTGGAAGCTGAA-3′(antisense) for FAIM2,5 ′-CAAGTGCGAGGGATGCAAAT-3′ (sense) and 5′-TGGCACACTTCTCACACTCC-3′(antisense) for MT3, 5′-CTCAGCCTAGAAGGGAATATCAAC-3′ (sense) and 5′-CAGCAGGGCTGATCCATTT-3′(antisense) for SH3GL2, and 5′-CACCCACTCCTCCACCTTTGA-3′ (sense) and 5′-ACCACCCTGTTGCTGTAGCCA-3′ (anti- sense) for GAPDH. The results were analyzed using the - ΔCT method with an unpaired t-test, and a P-value < 0.05 was considered a meaningful result.

Results

Screening of DEGs in MB

We downloaded the gene expression datasets—GSE42656, GSE109401, GSE50161, and GSE74195 from GEO. From the GSE42656 dataset, we identified 869 DEGs, of which 222 were upregulated, and 647 were downregulated. A total of 764 DEGs were identified from the GSE109401 data set—274 upregulated and 490 downregulated; 5,494 DEGs from GSE50161—2,798 upregulated and 2,696 downregulated, and 1,000 DEGs were identified from GSE74195—422 upregulated and 578 downregulated. Integrated analyses revealed that 104 DEGs were consistently expressed in the four data sets (Figs. 1A and 1B); these included 29 upregulated DEGs, and 75 downregulated DEGs in MB tissue compared to the normal brain tissue (Table 1). A heat map of the DEG distribution was generated using the GSE74195 dataset as a reference (Fig. S1).

Figure 1 Using the Venny map to obtain common DEGs in MB, and the cross areas represented the commonly DEGs of the four datasets (GSE42656, GSE109401, GSE50161, and GSE74195).

(A) Commonly up-regulated genes (29 DEGs); (B) Commonly down-regulated genes (75 DEGs).

Table 1 The 104 differentially expressed genes (DEGs), including 29 upregulated genes and 75 downregulated genes were identified in the medulloblastoma tissues from four profile datasets using normal brain tissues as a reference.

DEGs	Gene name	
Up-regulated 29	ACTL6A, UBE2C, TMEM97GTF2IRD1, INSM2, CDC20, PRC1, ZNF423, HMGB2, ODC1, MCM7, KLHDC8A, SOX11, RAD51AP1, KIF15, CDK6, EBF3, EYA2, TYMS, TTK, CD24, DACH1, SOX4, NUSAP1, RPGRIP1, KIF11, NEUROG1, OTX2, TOP2A	
Down-regulated 75	CADPS2, TMEM163, UNC13C, DNER, CNTNAP1, DNM3, TMOD1, FEZ1, RAPGEF4, RIT2, NDRG4, TPPP, TMEM55A, MT3, FAIM2, ABLIM1, RCAN2, MAP1A, DHCR24, NRXN2, PTGDS, CDS1, RASGRP1, PCP4, NRIP3, HPCAL4, PTCHD1, GAS7, KIAA0513, PMP2, PHACTR3, TF, CADM2, CNTN2, VSNL1, BCAS1, DKK3, SCRN1, FBXL16, ELOVL4, OPCML, DNM1, EEF1A2, TTC9B, CA11, CEND1, MOBP, SH3GL2, SCG5, EPDR1, LRRC3B, COX7A1, NAP1L3, TSPAN7, CAMK2B, OPTN, STMN4, CLSTN3, PEA15, SYN1, RNF175, REEP2, CSRP1, SYT11, TCEAL2, GPM6B, TPRG1L, ASTN1, MAP7D2, SIRPA, OLIG1, SYNGR3, ELMO1, HK1, VAMP2	
Notes.

Abbreviation DEGs differentially expressed genes

Gene Ontology Analysis of DEGs

The identified DEGs were analyzed using DAVID using p < 0.05 as the cut-off standard to identify the functions associated with the DEGs. The GO function annotation is divided into three functional groups—cell component (CC), molecular function (MF), and biological process (BP). As shown in Fig. 2 and Table 2, within the BPs the upregulated DEGs were closely related to mitotic cell cycle, cell cycle phase-M phase, cell cycle process, cell cycle, spindle organization, and microtubule-based process and the downregulated DEGs were closely related to vesicle-mediated transport. Within CC the upregulated DEGs were closely related to the spindle-microtubule, and cytoskeleton, whereas the downregulated DEGs were closely related to the synapse, synapse part, clathrin-coated vesicle, synaptic vesicle, cytoplasmic vesicle, coated vesicle, cytoplasmic membrane-bounded vesicle, vesicle, and membrane-bounded vesicle. Within MF, the upregulated DEGs showed a close relationship with DNA binding and the downregulated DEGs did not show any association.

Figure 2 Gene Ontology (GO) functional annotation analysis and significant terms of DEGs in MB and GO analysis divided into three parts.

Cyan for cell component, pink for molecular function, and blue for biological process.

Table 2 DEGs divided into upregulated genes and downregulated genes and their first ten meaningful GO enrichment analyses.

Term	Description	Count	P-Value	
Up-regulated				
GO:0005819	Spindle	6	1.92E–06	
GO:0000278	Mitotic cell cycle	8	5.99E–06	
GO:0003677	DNA binding	15	1.12E–05	
GO:0022403	Cell cycle phase	8	1.25E–05	
GO:0000279	M phase	7	3.80E–05	
GO:0022402	Cell cycle process	8	9.14E–05	
GO:0007049	Cell cycle	9	9.45E–05	
GO:0007051	Spindle organization	4	9.51E–05	
GO:0015630	Microtubule cytoskeleton	7	1.03E–04	
GO:0007017	Microtubule-based process	6	1.27E–04	
Down-regulated				
GO:0045202	Synapse	11	5.27E–06	
GO:0016192	Vesicle-mediated transport	11	5.69E–05	
GO:0044456	Synapse part	8	1.45E–04	
GO:0030136	Clathrin-coated vesicle	6	3.77E–04	
GO:0008021	Synaptic vesicle	5	4.37E–04	
GO:0031410	Cytoplasmic vesicle	11	7.37E–04	
GO:0030135	Coated vesicle	6	8.80E–04	
GO:0016023	Cytoplasmic membrane-bounded vesicle	10	9.86E–04	
GO:0031982	Vesicle	11	0.001022664	
GO:0031988	Membrane-bounded vesicle	10	0.001235649	
Notes.

Abbreviation GO Gene Ontology

DEGs differentially expressed genes

Figure 3 Kyoto Encyclopedia of Genes and Genomes (KEGG) signal pathway analysis of DEGs in MB.

Red represent the common up-regulated genes and green represent the common down-regulated genes.

KEGG pathway analysis

The 104 DEGs were grouped as upregulated and downregulated DEGs and the cellular signaling pathways represented were analyzed by KEGG (Fig. 3 and Table 3). The upregulated genes are mainly involved in the cell cycle, ubiquitin-mediated proteolysis, viral carcinogenesis, one-carbon pool by folate, and microRNAs in cancer. The downregulated genes are mainly involved in the synaptic vesicle cycle, bacterial invasion of epithelial cells, insulin secretion, HIF-1 signaling pathway, phospholipase D signaling pathway, and cell adhesion molecules (CAMs).

PPI network generation and identification of the hub gene

The interrelationships between the 104 DEGs were analyzed using STRING. We found that 100 of the 104 DEGs were related to each other and were visualized using Cytoscape –100 nodes and 776 edges are included in the network of PPI and the hub gene based on the degree of connectivity between genes were identified (Fig. 4A). Among the hub genes, the most significant hub genes were SYN1, CNTN2, FAIM2, MT3, and SH3GL2 (Fig. 4B).

Potential therapeutic drugs for MB

Of the 104 DEGs, only 85 were eventually converted to IDs from 22,214 probes in the Affymetrix platform. After that, we used CMap and identified six potential therapeutic drugs based on set criteria (Table 4). A PubMed literature search revealed that two drugs—vorinostat and resveratrol –have been reported to have a therapeutic effect in MB. The other three drugs, trichostatin A, pyrvinium, and prochlorperazine, have been reported to have therapeutic effects in other cancers. The 3D structures of these potential therapeutic drugs are available from PubChem (Fig. 5).

Experiments verify the five hub DEmRNAs in MB

To verify the expression levels of the five hub genes in the PPI network, RT-qPCR was used to detect their expression levels in normal human brain tissue and medulloblastoma cell lines (Daoy and D283). The five hub genes in the PPI network, including SYN1,CNTN2, FAIM2, MT3, and SH3GL2, were all down-regulated DEGs in MB, and this result was confirmed by RT-qPCR (Fig. 6). This shows that our analysis results were completely credible.

Table 3 KEGG pathway analysis of upregulated and downregulated genes in MB.

Database	ID	Term	Gene Count	P-Value	Gene names	
Up-regulated						
KEGG PATHWAY	hsa04110	Cell cycle	4	2.28E–06	TTK, CDC20, MCM7, CDK6	
KEGG PATHWAY	hsa04120	Ubiquitin mediated proteolysis	2	0.004624353	CDC20, UBE2C	
KEGG PATHWAY	hsa05203	Viral carcinogenesis	2	0.009970546	CDC20, CDK6	
KEGG PATHWAY	hsa00670	One carbon pool by folate	1	0.015198488	TYMS	
KEGG PATHWAY	hsa05206	MicroRNAs in cancer	2	0.020243092	CDK6, SOX4	
KEGG PATHWAY	hsa03030	DNA replication	1	0.026628344	MCM7	
KEGG PATHWAY	hsa03022	Basal transcription factors	1	0.033001239	GTF2IRD1	
KEGG PATHWAY	hsa00330	Arginine and proline metabolism	1	0.03652431	ODC1	
KEGG PATHWAY	hsa00480	Glutathione metabolism	1	0.037930065	ODC1	
KEGG PATHWAY	hsa05223	Non-small cell lung cancer	1	0.040735637	CDK6	
Down-regulated						
KEGG PATHWAY	hsa04721	Synaptic vesicle cycle	4	8.12E–06	DNM3, UNC13C, DNM1, VAMP2	
KEGG PATHWAY	hsa05100	Bacterial invasion of epithelial cells	3	0.000492126	DNM3, DNM1, ELMO1	
KEGG PATHWAY	hsa04911	Insulin secretion	3	0.000627009	RAPGEF4, CAMK2B, VAMP2	
KEGG PATHWAY	hsa04066	HIF-1 signaling pathway	3	0.001075892	HK1, TF, CAMK2B	
KEGG PATHWAY	hsa04072	Phospholipase D signaling pathway	3	0.002737401	DNM3, RAPGEF4, DNM1	
KEGG PATHWAY	hsa04514	Cell adhesion molecules (CAMs)	3	0.002843803	NRXN2, CNTN2, CNTNAP1	
KEGG PATHWAY	hsa04961	Endocrine and other Factor-regulated calcium reabsorption	2	0.003884233	DNM3, DNM1	
KEGG PATHWAY	hsa00524	Butirosin and neomycin biosynthesis	1	0.011244262	HK1	
KEGG PATHWAY	hsa04144	Endocytosis	3	0.01354801	DNM3, DNM1, SH3GL2	
KEGG PATHWAY	hsa04070	Phosphatidylinositol signaling system	2	0.0153706	CDS1, TMEM55A	
Notes.

Abbreviation KEGG Kyoto Encyclopedia of Genes and Genomes

ID identification

Discussion

Although several studies have been carried out to understand the pathogenesis of MB, there is no effective strategy to reduce the incidence or mortality of MB. This may be because most studies till date have focused on a single genetic event contributing to MB pathogenesis. Therefore, in order to identify effective molecular markers, we used data derived from four cohorts profile datasets obtained from the GEO database (GSE42656, GSE74195, GSE109401, GSE50161). Integrated analyses of the four data sets revelated 104 DEGs including 29 upregulated, and 75 downregulated DEGs. Enrichment of these genes identified certain cellular signaling pathways which may provide novel insights to the understanding of the pathogenesis of MB. We also used CMap, a drug development tool, and identified six potential drugs that can be used for the treatment of MB. To our knowledge, this is the first study to use integrated bioinformatical analysis for studying the pathogenesis of MB.

GO enrichment analysis showed that the upregulated DEGs were mainly involved in the spindle, cell cycle phase, M phase, cell cycle, spindle organization, microtubule cytoskeleton and the downregulated DEGs were mainly closely related to clathrin-coated vesicle, synaptic vesicle, and cytoplasmic vesicle. These results are consistent with previous literature reports that spindle defects play a vital role in the pathological process of MB (Abdelfattah et al., 2018). LCL161 an inhibitor of apoptosis proteins (IAP) combined with chemotherapy can slow MB cell proliferation by inducing G2/M phase arrest (Chen et al., 2018). Patupilone, a microtubule stabilizer, can reduce clonogenic survival and enhance the therapeutic efficacy of radiotherapy effect in MB (Oehler et al., 2011). α-synuclein binds to cytoplasmic vesicles to change the surface morphology of in U251 glioblastoma cells (Duan et al., 2017). Synaptic vesicle protein 2A is a predictor of the efficacy of levetiracetam in glioma patients (De Groot et al., 2011). The internalization of clathrin-coated vesicles can reduce the sensitivity of metabotropic receptors in C6 cells (Luis Albasanz, Fernandez & Martin, 2002).

Figure 4 The PPI network of DEGs.

(A) One hundred DEGs were incorporated into the network consisted of 29 upregulated (red) and 71 downregulated (green) genes; (B) hub genes determined by the degree of connectivity between DEGs.

KEGG pathway analysis demonstrated that the upregulated genes have significant enrichment in pathways including one-carbon pool by folate, microRNAs in cancer, DNA replication, and basal transcription factors. Previous studies have shown that these signaling pathways are mainly involved in the development of cancer. For example, the basal transcription factors such as oncogene Orthodenticle Homeobox 2, play an important role in cell migration and proliferation in MB (Wortham et al., 2014). Folate receptor (Folr1) participates in the pathway network of the one-carbon pool by folate, which is related to clinical, pathological, neuroimaging features, and prognosis of MB patients (Liu et al., 2017). MicroRNA-31 inhibits DNA replication by targeting minichromosome maintenance complex component 2 (MCM2), which has a strong inhibitory effect on MB growth (Jin et al., 2014). The downregulated genes were mainly involved in HIF-1 signaling pathway, phospholipase D signaling pathway, and phosphatidylinositol signaling system. These signaling pathways are involved in the pathological process of cancer. For example, phosphatidylinositol signaling system, such as the phosphatidylinositol 3-kinase (PI3K) pathway is associated with reduced survival in patients and is one of the most common signaling pathway-related abnormality in gliomas (Tuncel & Kalkan, 2018). Phospholipase D signaling pathway specifically inhibits autophagic flux and decreases GBM cell viability (Bruntz et al., 2014). The hypoxia-inducible factor-1 signaling pathway is vital for the invasion and metastasis of glioma cells (Yaghi et al., 2016). Together, our results show that the cellular pathways of DEGs identified in this study are closely related to the development of cancer.

Table 4 Six small-molecule compounds identified as potential drugs for MB treatment by CMap analysis.

Term	Enrichment	P-Value	
Vorinostat	−0.842	0	
Resveratrol	−0.827	0	
Trichostatin A	−0.568	0	
Pyrvinium	−0.727	0.00077	
Prochlorperazine	−0.475	0.00082	
0175029-0000	−0.718	0.00099	
Notes.

Abbreviation CMap connectivity map

Figure 5 The 3D structures of five small-molecule compounds identified as potential drugs for MB treatment.

(A) Vorinostat, (B) Resveratrol, (C) Trichostatin A, (D) Pyrvinium, and (E) Prochlorperazine.

Figure 6 The expression levels of the five hubgenes in the PP Inetwork were detected by RT-qPCR.

These included (A) SYN1, (B) CNTN2, (C) FAIM2, (D) MT3, and (E) SH3GL2 (P < 0.05).

The PPI network of DEGs represents an overview of their functional relations. Most of the hub genes selected in this network have been reported to be closely related to the pathological process of cancers. For example, SYN1, one of the neuronal genes, is related to the formation and maintenance of synaptic contacts and the expression of REST. RCOR1 may cause deregulated expression of SYN1, which contributes to the maintenance of glioblastoma stem-like cells (GSC) (Yucebas et al., 2016). CNTN2, a cell adhesion protein, is a downstream protein of RACK1, which affects the growth and differentiation of glioma cells through the RTK/Ras/MAPK signaling pathway (Yan & Jiang, 2016). FAIM2 is an anti-apoptotic molecule that promotes tumor cell growth through Fas-mediated mechanisms. Knock-down of FAIM2 can significantly affect tumor cell proliferation in small-cell lung cancer (Kang et al., 2016). MT3 is a tumor suppressor gene, and its expression is significantly reduced in AML samples. The overexpression of MT3 can inhibit cell proliferation and promote tumor cell apoptosis in pediatric acute myeloid leukemia (Tao et al., 2014). SH3GL2, as a suppressor for tumors, and has reduced expression in glioma tissues promoting migration and infiltration of glioma cells by enhancing STAT3/MMP2 signaling (Zhu et al., 2017). These five hub genes have important regulatory effects in the pathophysiology of cancers, but their role has not been reported in MB. These hub genes are potential targets for the treatment of MB.

CMap is an online tool that can be used to identify potential therapeutic drugs based on DEGs in various disease (Lamb et al., 2006). In the current study, we identified six drugs using CMap analysis. Two of these drugs have been previously reported to have a therapeutic effect on MB. The combination of vorinostat and MLN8237 can significantly inhibit the proliferation of MB cells (Muscal et al., 2013). Resveratrol inhibits the growth of cancer cells by regulating the Notch signaling pathway to promote apoptosis and differentiation of MB cells (Wang et al., 2008). Three drugs –trichostatin A, pyrvinium, and prochlorperazine –have not been reported for the treatment of MB; however, several studies have reported their treatment efficacies in other cancers. Prochlorperazine and trichostatin A were used for treating glioblastoma and pyrvinium for the treatment of ovarian cancer (Horing et al., 2013; Otreba & Buszman, 2018; Zhang et al., 2017). Although the six small-molecule compounds

obtained have not been adequately studied in MB. CMap is an online tool that provides researchers with an index based on disease DEGs to search for potential therapeutic drugs. Recently, some traditional drugs have been found to have new therapeutic effects. The development of new uses of old drugs can more quickly understand the pharmacological properties of drugs, which will help their early application in clinical treatment. For example, atorvastatin is a well-known traditional lipid-lowering drug, but in recent years, it has been found to have obvious therapeutic effects on chronic subdural hematoma (Jiang et al., 2018). Metformin is a traditional an anti-diabetic drug. In recent years, new therapeutic effects have been discovered in human diseases such as anti-tumor effects, neuroprotective effects, etc (Podhorecka, Ibanez & Dmoszynska, 2017; Saewanee et al., 2019). So we have reason to speculate that the small-molecule drugs identified by CMap analysis may be potential drugs for the treatment of MB.

Conclusions

In summary, most single data set analyses are limited by the small sample size, high experimental error, and lack of ethnic differences, and therefore cannot reliably identify the important genes and pathways involved in the pathogenesis of diseases. In this study, the above problems were well avoided by using multiple data sets and integrated analysis to improve the reliability and accuracy of the results. We identified 104 DEGs from four groups of gene expression profiling data of MB and analyzed them comprehensively. The hub genes identified by PPI network analysis include SYN1, CNTN2, FAIM2, MT3, and SH3GL2, which are involved in the pathogenesis of different cancers. We also identified several small-molecule compounds that may have potential therapeutic effects on MB. These findings provide new insights into the pathogenesis of MB and provide a basis for treatment. however, further experimental verification is needed.

Supplemental Information

Supplemental Information 1 The GSE74195 data set was used for a heat map to represent the common 104 DEGs

Red: upregulated; Green: downregulated

Click here for additional data file.

Additional Information and Declarations

Competing Interests

Author Contributions

Data Availability

The authors declare there are no competing interests.

Zhendong Liu and Ruotian Zhang conceived and designed the experiments, performed the experiments, analyzed the data, prepared figures and/or tables, authored or reviewed drafts of the paper, and approved the final draft.

Zhenying Sun performed the experiments, authored or reviewed drafts of the paper, and approved the final draft.

Jiawei Yao, Penglei Yao, Xin Chen, Xinzhuang Wang and Ming Gao analyzed the data, authored or reviewed drafts of the paper, and approved the final draft.

Jinzhao Wan and Yiming Du analyzed the data, prepared figures and/or tables, authored or reviewed drafts of the paper, and approved the final draft.

Shiguang Zhao conceived and designed the experiments, authored or reviewed drafts of the paper, and approved the final draft.

The following information was supplied regarding data availability:

The gene expression profile data is available at GEO: GSE42656, GSE74195, GSE109401, GSE50161.

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
