# Peer review of "Identification of hub genes and small-molecule compounds in medulloblastoma by integrated bioinformatic analyses"

_PeerJ, doi:10.7717/peerj.8670_

## Round 0.1 · original submission · Major Revisions

Dear Authors,

Major revisions are needed for this article as per the reviewer comments.

Thank you

Reviewer 1 ·

Basic reporting

The structure of article is well prepared, but the font is not consistent. Before starting experimental results, the rationale should be well addressed
In Introduction, Authors should summarize recent study consequence of gene expression profile in medulloblastoma. What is the unique point of the present study ??
All data in this manuscript is bioinformatics analysis. There is no any experimental confirmation. To strengthen the study quality, Authors may make more efforts on comparing their results with other published databases. Can Authors provide evidence to conclude which signaling is major to promote medullablastoma ??

Experimental design

It is appreciated that all data used for bioinformatic analysis is available, and methods were clearly described. The rationale for each experiment should be described in the beginning of results.

Validity of the findings

Actually, numerous studies are attempting to establish gene expression databases for precision medicine. In addition, most of gene expression data are available online. Hence, the novelty of this study is intensively limited.
The current study will get improved if the confirmation for gene expression in some human specimens is performed.

Reviewer 2 ·

Basic reporting

The article is well written, fluent and well structured in all its part (introduction, methods, results, discussion and conclusion)

Experimental design

The aim, methods and limitations are clearly defined.
Results are relevant and deeply commented in the discussion.

Validity of the findings

Meaningful results. I would stress the gap that still exist between these results and treatments although a better knowledge of genetic basis could reduce the gap.

Additional comments

Good paper. Nothing too add.

·

Basic reporting

The focus of this paper is the bioinformatic analysis of RNA microarray expression in medulloblastoma (MB), the most common brain cancer in children. The goal was the identification of differentially expressed genes. After expression analysis, the authors also performed in silico identification of potential new molecules for personalized target therapy of MB.
The authors started their study from microarray expression data deposited in a public NCBI database, Gene Expression Omnibus database (GEO). From this database, they have selected four different datasets derived from four published studies. Indeed, in GEO there are many deposited datasets relative to MB microarray. It is not particularly evident how and why the authors made their choice. The authors should explain better.
From this starting point, the authors performed an excellent bioinformatic analysis. The paper is very well written with clear, unambiguous, technically correct terms. The article includes a sufficient introduction and background to demonstrate how the work fits into the broader field of knowledge. The study is very well performed following a pathway that is becoming very common for expression data analysis in cancer. A paper that uses a very similar approach to study RNA expression data in MB was recently published (Fateme Shaabanpour Aghamaleki, Behrouz Mollashahi , Nika Aghamohammadi , Nematollah Rostami , Zeinab Mazloumi , Hamidreza Mirzaei, Afshin Moradi, Mojgan Sheikhpour , Abolfazl Movafagh. Bioinformatics Analysis of Key Genes and Pathways for Medulloblastoma as a Therapeutic Target in Asian Pacific Journal of Cancer Prevention, vol 20, page 221, 2019). The authors do not mention this paper and, although their manuscript is undoubtedly superior in the analysis and original in the approach to use CMap software to identify small molecule compounds as effective drugs for MB, they should discuss the differences and the similarities with the published literature. Relevant prior research should be appropriately referenced.
The structure of the article is in an acceptable format; the figures are relevant to the content of the article, of sufficient resolution, and appropriately described and labeled. All appropriate raw data have been made available under the journal policy.

Experimental design

The study is within the Aims and Scope of the journal. The paper defines clearly the research question, which is relevant and meaningful. The investigation was conducted rigorously and with a high technical standard. Methods are described with sufficient details.

Validity of the findings

The data are robust, bioinformatically sound, and controlled. The impact of the findings is undetermined. The degree of advance is not immediately apparent. The molecules identified by CMap analysis are a potential novelty that the authors should discuss and stress better. Also, the conclusions are well stated and related to the original question investigated.

Additional comments

The authors have done an excellent and smart bioinformatic analysis of microarray expression data of MB. They used their analysis to identify in silico new compounds to test in MB therapy with a new intelligent approach. They should better discuss their interesting data and compare them to published literature to deserve publication

---

## Round 0.2 · accepted · Accept

Dear Authors,

Congratulations on your accepted manuscript.

Reviewer 1 ·

Basic reporting

All questions and requests were answered well. I have no further questions.

Experimental design

Clear

Validity of the findings

It is benefited after confirmation by qPCR.

Additional comments

No further requests

·

Basic reporting

The authors can now show the robustness of their in silico data analysis by in vivo RT-qPCR measurements that experimentally confirm the downregulation of 5 hub genes. I believe the manuscript is now in better shape and can be considered for publication.

Experimental design

The authors can now show the robustness of their in silico data analysis by in vivo RT-qPCR measurements that experimentally confirm the downregulation of 5 hub genes. I believe the manuscript is now in better shape and can be considered for publication.

Validity of the findings

The authors can now show the robustness of their in silico data analysis by in vivo RT-qPCR measurements that experimentally confirm the downregulation of 5 hub genes. I believe the manuscript is now in better shape and can be considered for publication.

Additional comments

The authors can now show the robustness of their in silico data analysis by in vivo RT-qPCR measurements that experimentally confirm the downregulation of 5 hub genes. I believe the manuscript is now in better shape and can be considered for publication.